# A Solver-Free Framework for Scalable Learning in Neural ILP Architectures

**Yatin Nandwani**,* **Rishabh Ranjan**,* **Mausam & Parag Singla**
Department of Computer Science, Indian Institute of Technology Delhi, INDIA
{yatin.nandwani, rishabh.ranjan.cs118, mausam, parags}@cse.iitd.ac.in

## Abstract

There is a recent focus on designing architectures that have an Integer Linear Programming (ILP) layer following a neural model (referred to as *Neural ILP* in this paper). Neural ILP architectures are suitable for pure reasoning tasks that require data-driven constraint learning or for tasks requiring both perception (neural) and reasoning (ILP). A recent SOTA approach for end-to-end training of Neural ILP explicitly defines gradients through the ILP black box [Paulus et al., 2021] – this trains extremely slowly, owing to a call to the underlying ILP solver for every training data point in a minibatch. In response, we present an alternative training strategy that is *solver-free*, i.e., does not call the ILP solver at all at training time. Neural ILP has a set of trainable hyperplanes (for cost and constraints in ILP), together representing a polyhedron. Our key idea is that the training loss should impose that the final polyhedron separates the positives (all constraints satisfied) from the negatives (at least one violated constraint or a suboptimal cost value), via a soft-margin formulation. While positive example(s) are provided as part of the training data, we devise novel techniques for generating negative samples. Our solution is flexible enough to handle equality as well as inequality constraints. Experiments on several problems, both perceptual as well as symbolic, which require learning the constraints of an ILP, show that our approach has superior performance and scales much better compared to purely neural baselines and other state-of-the-art models that require solver-based training. In particular, we are able to obtain excellent performance on $9 \times 9$ symbolic and visual sudoku, to which the other Neural ILP model is not able to scale. [2]

## 1 Introduction

There has been a growing interest in the community which focuses on developing neural models for solving combinatorial optimization problems. These problems often require complex reasoning over discrete symbols. Many of these problems can be expressed in the form an underlying Integer Linear Program (ILP). Two different kinds of problem (input) settings have been considered in the literature: (a) purely symbolic and (b) combination of perceptual and symbolic. Solving an n-Queens problem given the partial assignment of queens on the board as input would be an example of the former, and solving a sudoku puzzle given the image of a partially filled board as input would be an example of the latter. While the first setting corresponds to a pure reasoning task, second involves a combination of perception and reasoning tasks which need to be solved in a joint fashion. Existing literature proposes various approaches to handle one or both these settings. One line of work proposes purely neural models [Palm et al., 2018, Nandwani et al., 2021, Dong et al., 2019] for solving these tasks representing the underlying constraints and costs implicitly. While standard CNNs are used to

---

*Equal contribution. Work done while at IIT Delhi. Current email: rishabhr@andrew.cmu.edu
[2]Code available at: *https://github.com/dair-iitd/ilploss*

solve the perceptual task, neural models such as Graph Neural Networks (GNNs) take care of the reasoning component. In an alternate view, one may want to solve these tasks by explicitly learning the constraints and cost of the underlying ILP. While the perceptual reasoning would still be handled by using modules such as CNN, reasoning is taken care of by having an explicit representation in the form of an ILP layer representing constraints and costs. Such an approach would have the potential advantage of being more interpretable, and also being more accurate if the underlying constraints could be learned in a precise manner. Some recent approaches take this route, and include works by [Paulus et al., 2021, Pogancic et al., 2020, Berthet et al., 2020]. We refer to the latter set of approaches as *Neural ILP* architectures. [3]

Learning in Neural ILP architectures is complicated by the fact that there is a discrete optimization (in the form of an ILP layer) at the end of the network, which is typically non-differentiable, making the end-to-end learning of the system difficult. One of the possible ways is to instead use an iterative ILP solving algorithm such as a cutting-plane method [Gomory, 2010] that uses a continuous relaxation in each iteration which is shown to be differentiable due to the introduction of continuous variables [Ferber et al., 2020, Wilder et al., 2019]. Most of these works are concerned with learning only the cost function and assume that the constraints are given. Recent work by Paulus et al. [2021] has proposed an approach to directly pass the gradients through a black-box ILP solver. Specifically, they rely on Euclidean distances between the constraint hyperplanes and the current solution obtained by the ILP solver to produce gradients for backprop. Though this approach improves the quality of results compared to earlier works involving continuous relaxations, scalability gets severely affected since an ILP has to be solved at every learning iteration, making the training extremely slow.

We are interested in answering the following question: is there a way to train a neural ILP architecture in an end-to-end manner, which does not require access to an underlying solver? Such an approach, if exists, could presumably result in significantly faster training times, resulting in scalability. In response, we propose a novel technique to back-propagate through the learnable constraints as well as the learnable cost function of an unknown Integer Linear Program (ILP). During training, our technique doesn't solve the ILP to compute the gradients. Instead, we cast the learning of ILP constraints (and cost) as learning of a polyhedron, consisting of a set of hyperplanes, such that points inside the polyhedron are treated as positive, and points outside as negative. While a positive point needs to be classified as positive by each of the hyperplanes, a negative point needs to be classified as negative only by at least one of the hyperplanes. Our formulation incorporates the learning of ILP cost also as learning of one of the hyperplanes in the system. We formulate a novel margin-based loss to learn these hyperplanes in a joint fashion. A covariance based regularizer that minimizes the cosine similarity between all pairs of hyperplanes ensures that learned constraints (hyperplanes) are not redundant. Since the training data comes only with positive examples, i.e., solutions to respective optimization problems, we develop several techniques for sampling the negatives, each of which is central to the effective learning of our hyperplanes in our formulation.

We present several experiments on problems which require learning of ILP constraints and cost, with both symbolic as well as perceptual input. These include solving a symbolic sudoku as well as visual sudoku in which we are given an image of a partially filled board [Wang et al., 2019] (perceptual); ILPs with random constraints (symbolic), Knapsack from sentence description (perceptual) and key-point matching (perceptual) from [Paulus et al., 2021]. Our closest competitor, CombOptNet [Paulus et al., 2021], can not solve even the smallest of the sudoku boards sized $4 \times 4$, whereas we can easily scale to $9 \times 9$, getting close to 100% accuracy. We are slightly better on key-point matching, and obtain significantly better accuracy on random ILPs and knapsack, especially on large problem sizes. We also outperform purely neural baselines (wherever applicable).

## 2 Related Work

When the constraints of an ILP are known, many works propose converting such constraints over discrete output variables into equivalent constraints over their probabilities given by the model and backpropagate over them [Nandwani et al., 2019, Li and Srikumar, 2019]. Some works use an ILP solver with the known constraints during inference to improve the predictions [Ning et al., 2019, Han et al., 2019]. When constraints have to be learned, which is the focus of this paper, one line of work uses a neural model, such as a GNN, to replace a combinatorial solver altogether and *implicitly*

---

[3]Although these methods can also train a Neural-ILP-Neural architecture, studying this is beyond our scope.

encodes the constraints in model's weights and learns them from data. Nandwani et al. [2022, 2021], Palm et al. [2018] train a Recurrent Relational Network for learning to solve puzzles like sudoku, graph coloring, Futoshiki, etc; Dong et al. [2019] propose Neural Logic Machines (NLMs) that learn lifted first order rules and experiment with blocks world, reasoning on family trees etc; Ranjan et al. [2022] use a siamese GNN architecture to learn the combinatorial problems of graph and subgraph distance computation; Bajpai et al. [2018], Garg et al. [2019, 2020], Sharma et al. [2022] use GNNs to train probabilistic planners for combinatorial domains with large, discrete state and action spaces; Selsam et al. [2019], Amizadeh et al. [2019a,b] train a GNN for solving any CSP when constraints are *explicitly* given in a standard form such as CNF, DNF or Boolean Circuits. This is clearly different from us since we are interested in explicitly learning the constraints of the optimization problem.

Inverse Optimization [Chan et al., 2021] aims to learn the constraints (cost) of a linear program (LP) from the observed optimal decisions. Recently [Tan et al., 2019, 2020] use the notion of Parameterized Linear Programs (PLP) in which both the cost and the constraints of an LP are parameterized by unknown weights. [Gould et al., 2019] show how to differentiate through continuous constrained optimization problem using the notion of a '*declarative node*' that uses implicit function theorem [Mingari Scarpello and Ritelli, 2002]. Similar to this are other works [Amos and Kolter, 2017, Agrawal et al., 2019] that define methods for differentiating through other continuous problems like convex quadratic programs (QPs) or cone programs. While all these techniques are concerned with learning the constraints (cost) for optimization problems defined over continuous variables, our focus is on learning constraints (cost) for an ILP which involves optimization over discrete variables and can be a significantly harder problem.

In the space of learning cost for an ILP, several approaches have been proposed recently. [Pogancic et al., 2020] give the gradient w.r.t. linear cost function that is optimized over a given discrete space. [Rolínek et al., 2020a,b] exploit it for the task of Deep Graph Matching, and for differentiating through rank based metrics such as Average Precision/Recall respectively. [Berthet et al., 2020] replace the black-box discrete solver with its smooth perturbed version during training and exploit 'perturb-and-MAP' [Papandreou and Yuille, 2011] to compute the gradients. On similar lines, [Niepert et al., 2021] also exploit perturb-and-MAP but propose a different noise model for perturbation and also extends to the case when the output of the solver may feed into a downstream neural model. These methods assume the constraints to be given, and only learn the cost for an ILP.

Other methods use relaxations of specific algorithms for the task of learning the constraints and cost for an ILP. Ferber et al. [2020] backprop through the KKT conditions of the LP relaxation created iteratively while using cutting-plane method [Gomory, 2010] for solving an MILP; Wilder et al. [2019] add a quadratic penalty term to the continuous relaxation and use implicit function theorem to backprop through the KKT conditions, as done in Amos and Kolter [2017] for smooth programs; Mandi and Guns [2020] instead add a log-barrier term to get the LP relaxation and differentiate through its homogeneous self-dual formulation linking it to the iterative steps in the interior point method. Wang et al. [2019] uses a low rank SDP relaxation of MAXSAT and defines how the gradients flow during backpropagation. Presumably, these approaches are limited in their application since they rely on specific algorithmic relaxations.

Instead of working with a specific algorithm, some recent works differentiate through a black-box combinatorial solver for the task of learning constraints [Paulus et al., 2021]. The solver is called at every learning step to compute the gradient. This becomes a bottleneck during training, since the constraints are being learned on the go, and the problem could become ill-behaved resulting in a larger solver time, severely limiting scalability of such approaches. In contrast, we would like to perform this task in a solver-free manner. Pan et al. [2020] propose learning of constraints for the task of structured prediction. They represent the linear constraints compactly using a specific two layer Rectifier Network [Pan and Srikumar, 2016]. A significant limitation is that their method can only be used for learning constraints, and not costs. Further, their approach does not do well experimentally on our benchmark domains. Meng and Chang [2021] propose a non-learning based approach for mining constraints in which for each training data $(\mathbf{c}, \mathbf{y})$, a new constraint is added which essentially imply that no other target can have better cost than the given target $\mathbf{y}$ for the corresponding cost $\mathbf{c}$. We are specifically interested in learning not only the constraints, but also the cost of an ILP from data. Convex Polytope Machines (CPM) [Kantchelian et al., 2014] learns a non linear binary classifier in the form of a polytope from a dataset of positive and negative examples. In contrast, our goal is to learn the cost as well as constraints for an ILP where both the constraints and the cost could be parameterized by another neural network. Also, we do not have access to any negative samples.

# 3 Differentiable ILP Loss

## 3.1 Background and Task Description

We are interested in learning how to solve combinatorial optimization problems that can be expressed as an Integer Linear Program (ILP) with equality as well as inequality constraints:

$$\arg\min_{\mathbf{z}\in\mathbb{Z}^n} \mathbf{c}^T\mathbf{z} \text{ subject to } \mathbf{Uz} = \mathbf{v}; \ \mathbf{Gz} + \mathbf{h} \geq \mathbf{0} \tag{1}$$

Here $\mathbf{c} \in \mathbb{R}^n$ represents an $n$ dimensional cost vector, the matrix $\mathbf{U} \in \mathbb{R}^{m_1 \times n}$ and $\mathbf{v} \in \mathbb{R}^{m_1}$ represent $m_1$ linear equality constraints, and the matrix $\mathbf{G} \in \mathbb{R}^{m_2 \times n}$ and $\mathbf{h} \in \mathbb{R}^{m_2}$ represent $m_2$ linear inequality constraints, together defining the feasible region.

Without loss of generality, one can replace an equality constraint $\mathbf{u}_i^T\mathbf{z} = \mathbf{v}_i$ by two inequality constraints: $\mathbf{u}_i^T\mathbf{z} \geq \mathbf{v}_i$ and $-\mathbf{u}_i^T\mathbf{z} \geq -\mathbf{v}_i$. Here $\mathbf{u}_i$ and $\mathbf{v}_i$ represent the $i^{th}$ row of the matrix $\mathbf{U}$ and $i^{th}$ element of the vector $\mathbf{v}$ respectively. Using this, one can reduce ILP in eq. (1) to an equivalent form with just inequality constraints:

$$\arg\min_{\mathbf{z}\in\mathbb{Z}^n} \mathbf{c}^T\mathbf{z} \text{ subject to } \mathbf{Az} + \mathbf{b} \geq \mathbf{0} \tag{2}$$

Here matrix $\mathbf{A} \in \mathbb{R}^{m \times n}$ is row-wise concatenation of the matrices $\mathbf{G}, \mathbf{U}$ and $-\mathbf{U}$. Vector $\mathbf{b} \in \mathbb{R}^m$ is a row-wise concatenation of vectors $\mathbf{h}, -\mathbf{v}$ and $\mathbf{v}$, and $m = 2m_1 + m_2$ is the total number of inequality constraints. We represent the $i^{th}$ constraint $\mathbf{a}_i^T\mathbf{z} + \mathbf{b}_i \geq 0$, as $[\mathbf{a}_i|\mathbf{b}_i]$. The integer constraints $\mathbf{z} \in \mathbb{Z}^n$ make the problem NP hard.

**Neural ILP Architecture:** In a neural ILP architecture, the constraint matrix $\mathbf{A}$, vector $\mathbf{b}$, and the cost vector $\mathbf{c}$ are neural functions $f_\mathbf{A}, f_\mathbf{b}$, and $f_\mathbf{c}$ (of input $x$) parameterized by learnable $\Theta = (\theta_\mathbf{A}, \theta_\mathbf{b}, \theta_\mathbf{c})$, i.e., $\mathbf{A} = f_\mathbf{A}(x; \theta_\mathbf{A})$; $\mathbf{b} = f_\mathbf{b}(x; \theta_\mathbf{b})$; $\mathbf{c} = f_\mathbf{c}(x; \theta_\mathbf{c})$. This results in a different ILP for each input $x$. For an input $x$, the solution given by a neural ILP model, $\mathbf{M}_\Theta(x)$, is nothing but the optima of the following paramterized ILP where $\mathbf{A}, \mathbf{b}, \mathbf{c}$ are replaced by corresponding neural functions $f_\mathbf{A}, f_\mathbf{b}, f_\mathbf{c}$ evaluated at input $x$:

$$\mathbf{M}_\Theta(x) = \arg\min_{\mathbf{z}} (f_\mathbf{c}(x; \theta_\mathbf{c}))^T \mathbf{z} \text{ subject to } f_\mathbf{A}(x; \theta_\mathbf{A})\mathbf{z} + f_\mathbf{b}(x; \theta_\mathbf{b}) \geq \mathbf{0}, \mathbf{z} \in \mathbb{Z}^n \tag{3}$$

**Example of visual-sudoku:** In $k \times k$ visual-sudoku, the input $x$ is an image of a sudoku puzzle and $\mathbf{y}^* \in \{0,1\}^n$ is the corresponding solution, represented as a $n = k^3$ dimensional binary vector: the integer in each of the $k^2$ cells is represented as a $k$ dimensional one-hot binary vector. Function $f_\mathbf{c}(x; \theta_\mathbf{c})$ parameterizing the cost is nothing but a neural digit classifier that classifies the content of each of the $k^2$ cells into one of the $k$ classes. The neural functions $f_\mathbf{A}$ and $f_\mathbf{b}$ are independent of the input $x$ as the constraints are the same for every $k \times k$ sudoku puzzle. Therefore, $f_\mathbf{A}(x; \theta_\mathbf{A}) = \theta_\mathbf{A}$, and $f_\mathbf{b}(x; \theta_\mathbf{b}) = \theta_\mathbf{b}$, where $\theta_\mathbf{A}$ is just a learnable matrix of dimension $m \times n$ and $\theta_\mathbf{b}$ is a learnable vector of dimension $m$. See Bartlett et al. [2008] for an ILP formulation of $k \times k$ sudoku with $4k^2$ equality constraints in a $n = k^3$ dimensional binary space $\{0,1\}^n$.

**Learning Task:** Given a training dataset $\mathcal{D} = \{(x_s, \mathbf{y}_s^*) \mid s \in \{1 \dots S\}\}$ with $S$ training samples, the task is to learn the parameters $\Theta = (\theta_\mathbf{A}, \theta_\mathbf{b}, \theta_\mathbf{c})$, such that $\mathbf{y}_s^* = \mathbf{M}_\Theta(x_s)$ for each $s$. To do so, one needs to define derivatives $\frac{\partial L}{\partial \mathbf{A}}, \frac{\partial L}{\partial \mathbf{b}}$, and $\frac{\partial L}{\partial \mathbf{c}}$ w.r.t. $\mathbf{A}, \mathbf{b}$, and $\mathbf{c}$ respectively, of an appropriate loss function $L$. Once such a derivative is defined, one can easily compute derivatives w.r.t.$\theta_\mathbf{A}, \theta_\mathbf{b}$, and $\theta_\mathbf{c}$ using the chain rule: $\frac{\partial L}{\partial \theta_\mathbf{A}} = \frac{\partial L}{\partial \mathbf{A}} \frac{\partial f_\mathbf{A}(x; \theta_\mathbf{A})}{\partial \theta_\mathbf{A}}$; $\frac{\partial L}{\partial \theta_\mathbf{b}} = \frac{\partial L}{\partial \mathbf{b}} \frac{\partial f_\mathbf{b}(x; \theta_\mathbf{b})}{\partial \theta_\mathbf{b}}$; and $\frac{\partial L}{\partial \theta_\mathbf{c}} = \frac{\partial L}{\partial \mathbf{c}} \frac{\partial f_\mathbf{A}(x; \theta_\mathbf{c})}{\partial \theta_\mathbf{c}}$. Hence, in the formulation below, we only worry about computing gradients w.r.t. the constraint matrix $\mathbf{A}$, vector $\mathbf{b}$, and the cost vector $\mathbf{c}$.

The existing approaches, e.g., [Paulus et al., 2021], explicitly need access to current model prediction $\mathbf{M}_\Theta(x)$. This requires solving the ILP in eq. (3), making the learning process extremely slow. In contrast, we present a *'solver-free'* framework for computation of an appropriate loss and its derivatives w.r.t. $\mathbf{A}, \mathbf{b}$, and $\mathbf{c}$. Our framework does not require solving any ILP while training, thereby making it extremely scalable as compared to existing approaches.

## 3.2 A Solver-free Framework

**Conversion to a constraint satisfaction problem:** As a first step, we convert the constraint optimization problem in eq. (2) to an equivalent constraint satisfaction problem by introducing an additional

'*cost–constraint*': $\mathbf{c}^T \mathbf{z} \leq \mathbf{c}^T \mathbf{y}^*$, equivalent to $\mathbf{a}_{m+1} = -\mathbf{c}$ and $\mathbf{b}_{m+1} = \mathbf{c}^T \mathbf{y}^*$.

Note that the above cost–constraint separates the solution $\mathbf{y}^*$ of the original ILP in eq. (2) from the other feasible integral points. This is because $\mathbf{y}^*$ must achieve the minimum objective value $\mathbf{c}^T \mathbf{y}^*$ amongst all the feasible integral points and hence no other feasible integral point $\mathbf{z}$ can obtain objective value less than or equal to $\mathbf{c}^T \mathbf{y}^*$. The new constraint $[\mathbf{a}_{m+1}|\mathbf{b}_{m+1}]$ together with the original $m$ constraints guarantee that $\mathbf{y}^*$ is the only solution satisfying all of the $(m+1)$ constraints. This results in the following equivalent linear constraint satisfaction problem:

$$\underset{\mathbf{z} \in \mathbb{Z}^n}{\arg\min} \, \mathbf{0}^T \mathbf{z} \text{ subject to } \mathbf{A}\mathbf{z} + \mathbf{b} \geq \mathbf{0} \; ; \; \mathbf{c}^T \mathbf{z} \leq \mathbf{c}^T \mathbf{y}^* \tag{4}$$

Constructing such an equivalent satisfaction problem requires access to the solution $\mathbf{y}^*$ of the original ILP in eq. (2) and that is already available to us for the training data samples. By rolling up the cost vector $\mathbf{c}$ into an additional constraint with the help of $\mathbf{y}^*$, we have converted our original objective of learning both the cost and constraints to just learning of constraints in eq. (4).

The main intuition behind our framework comes from the observation that each of the $m$ linear constraints defining the feasible region are essentially $m$ linear binary classifiers (hyperplanes) in $\mathbb{R}^n$ separating the ground truth $\mathbf{y}^*$ from the infeasible region. The additional cost–constraint in eq. (4) separates $\mathbf{y}^*$ from other integral samples feasible *w.r.t.* the original $m$ constraints. Learning constraints of an ILP is akin to simultaneously learning $m$ linear binary classifiers separating $\mathbf{y}^*$ from other infeasible points of the original ILP along with learning a classifier for cost–constraint, separating $\mathbf{y}^*$ from other feasible integral points.

For a vector $\mathbf{y}^*$ to lie inside the feasible region, *all* the classifiers need to classify it positively, and hence it acts as a positive data point for all the $m+1$ binary classifiers. In the absence of explicitly provided negative samples, we propose a couple of strategies for sampling them from $\mathbb{Z}^n$ for each ground truth $\mathbf{y}_s^*$. We discuss them in detail in section 3.3. For now, let $\mathcal{N}_{\mathbf{y}^*}$ be the set of all the negative samples generated by all our sampling techniques for a ground truth positive sample $\mathbf{y}^*$. In contrast to a positive point which needs to satisfy *all* the $m+1$ constraints, a negative point becomes infeasible even if it violates *any one* of the $m+1$ constraints. As a result, it would suffice if any one of the $m+1$ classifiers correctly classify it as a negative sample (refer to appendix for an illustration). While learning, one should not assign a negative sample to any specific classifier as their parameters are being updated continuously. Instead, we make a soft assignment depending upon the distance of the negative sample from the hyperplane. With this intuition, we now formally define our ILP–Loss.

**Formulating solver-free ILP–Loss:** Let $d(\mathbf{z}; [\mathbf{a}_i|\mathbf{b}_i]) = \frac{\mathbf{a}_i^T \mathbf{z} + \mathbf{b}_i}{|\mathbf{a}_i|}$ represent the signed Euclidean distance of a point $\mathbf{z} \in \mathbb{R}^n$ from the hyperplane corresponding to the constraint $[\mathbf{a}_i|\mathbf{b}_i]$. We want the signed distance from all the hyperplanes to be positive for the ground truth samples and negative from at least one hyperplane for all the sampled negative points. We operationalize this via a margin based loss function:

$$L(\mathbf{A}, \mathbf{b}, \mathbf{c}, \mathbf{y}^*|\mathcal{N}_{\mathbf{y}^*}) \;=\; \lambda_{pos} L_+ + \lambda_{neg} L_- + \lambda_{cov} L_o \quad \text{where} \tag{5}$$

$$L_+ \;=\; \frac{1}{m} \sum_{i=1}^{m} \max\{0, \mu^+ - d(\mathbf{y}^*; [\mathbf{a}_i|\mathbf{b}_i])\} \tag{6}$$

$$L_- \;=\; \frac{1}{|\mathcal{N}_{\mathbf{y}^*}|} \sum_{\mathbf{y}^- \in \mathcal{N}_{\mathbf{y}^*}} \sum_{i=1}^{m+1} w_{\mathbf{y}^-}^i \max\{0, \mu^- + d(\mathbf{y}^-; [\mathbf{a}_i|\mathbf{b}_i])\} \tag{7}$$

$$L_o \;=\; \sum_{i,j=1; i \neq j}^{m} \frac{\mathbf{a}_i^T \mathbf{a}_j}{|\mathbf{a}_i||\mathbf{a}_j|} \;\simeq\; \left( \sum_{i=1}^{m} \frac{\mathbf{a}_i}{|\mathbf{a}_i|} \right)^2 \tag{8}$$

$$w_{\mathbf{y}^-}^i \;=\; \frac{e^{\left(-d(\mathbf{y}^-; [\mathbf{a}_i|\mathbf{b}_i])/\tau\right)}}{\sum\limits_{j=1}^{m+1} e^{\left(-d(\mathbf{y}^-; [\mathbf{a}_j|\mathbf{b}_j])/\tau\right)}} \tag{9}$$

We call $L_+$, $L_-$, and $L_o$ as the positive, negative and covariance loss respectively. The average in $L_+$ ensures that a ground truth sample is positively classified by *all* the classifiers. $\mu^+$ and $\mu^-$ are the hyperparameters representing the margins for the positive and the negative points respectively. $w_{\mathbf{y}^-}^i$ in $L_-$ represents the soft assignment of $\mathbf{y}^-$ to the $i^{th}$ constraint $[\mathbf{a}_i|\mathbf{b}_i]$ and is computed by temperature

annealed softmax over the negative distances in eq. (9). Softmax ensures that the hyperplane which is most confident to classify it as negative gets the maximum weight. When $\mathbf{y}^-$ lies inside the feasible region, then the most confident classifier is the one closest to $\mathbf{y}^-$. To avoid the pathological behaviour of decreasing the loss by changing the weights, we ensure that gradients do not flow through $w_{\mathbf{y}^-}^i$ in eq. (7).

The temperature parameter $\tau$ needs to be annealed as the training progresses. A high temperature initially can be seen as '*exploration*' for the right constraint that will be violated by $\mathbf{y}^-$. This is important as the constraints are also being learnt and are almost random initially, so a given negative $\mathbf{y}^-$ should not commit to a particular hyperplane. Additionally, this encourages multiple constraints to be violated for each negative, which leads to a robust set of constraints. As the training progresses, we reduce the temperature $\tau$, ensuring that the most confident classifier with the least signed distance gets almost all the weight, which can be seen as '*exploitation*' of the most confident classifier. If $\mathbf{y}^-$ is correctly classified as a negative with a margin $\mu^-$ by *any* classifier *i.e.*, $d(\mathbf{y}^-; [\mathbf{a}_i|\mathbf{b}_i]) \leq -\mu^-$ for some $i$, then the corresponding negative loss, $\max\{0, \mu^- + d(\mathbf{y}^-; [\mathbf{a}_i|\mathbf{b}_i])\}$, becomes zero, and a low value of $\tau$ ensures that it gets all the weight.

The last term $L_o$ acts as a regularizer and tries to ensure that no two learnt constraints are similar. We call it the *covariance loss* as it maximizes the covariance between the constraint unit vectors. Equivalently, it minimizes the cosine similarity between all pairs of constraints. The weights $\lambda_{pos}$, $\lambda_{neg}$, and $\lambda_{cov}$ are computed dynamically during training with a multi-loss weighing technique using coefficient of variations as described in Groenendijk et al. [2021]. Intuitively, the loss term with maximum variance over the learning iterations adaptively gets most of the weight.

**Other details:**

**Parameterization of equality constraints:** Recall that we replace an equality $\mathbf{u}_i^T \mathbf{z} = \mathbf{v}_i$ by two inequality constraints: $\mathbf{v}_i \leq \mathbf{u}_i^T \mathbf{z} \leq \mathbf{v}_i$. In practice, to enhance learnability, we add a small margin of $\epsilon$ on both sides: $\mathbf{v}_i - \epsilon \leq \mathbf{u}_i^T \mathbf{z} \leq \mathbf{v}_i + \epsilon$. We pick an $\epsilon$ small enough so that the probability of the new feasible region to include an infeasible integral point is negligible, but higher than $\mu^+$ so that the positive point can be inside the polyhedron by the specified margin.

**Known boundaries:** In many cases, the boundary conditions on the output variables are known, *i.e.*, $\mathbf{z} \in \mathcal{Y} = [l, u]^n$, where $l$ and $u$ are the lower and upper bounds on each dimension of $\mathbf{z}$. We handle this by adding known boundary constraints: $l \leq \mathbf{z}_i \leq u, \forall i \in \{1 \cdots n\}$.

**Over-parameterization of constraints:** As done in Paulus et al., we also over-parameterize each constraint hyperplane by an additional learnable offset vector $\mathbf{o}_i$ which can be viewed as its own local origin. Radius $\mathbf{r}_i$ represents its distance from its own origin $\mathbf{o}_i$, resulting in the following hyperplane in the base coordinate system: $\mathbf{a}_i^T \mathbf{z} + \mathbf{r}_i - \mathbf{o}_i^T \mathbf{a}_i / |\mathbf{a}_i| \geq 0$.

**Initialization of A:** The way we initialize the constraints may have an impact on the learnability. While [Paulus et al., 2021] propose to sample each entry of $\mathbf{a}_i$ uniformly (and independently) between $[-0.5, 0.5]$, we also experiment with a standard Gaussian initialization. The latter results in initial hyperplanes with their normal directions ($\mathbf{a}_i$'s) uniformly sampled from a unit hyper-sphere. In expectation, such initialization achieves minima of $L_o$ that measures total pairwise covariance.

### 3.3 Negative Sampling

A meaningful computation of ILP–Loss in eq. (5) depends crucially on the negative samples $\mathcal{N}_{\mathbf{y}}^*$. Randomly sampling negatives from $\mathbb{Z}^n$ is one plausible strategy, but it may not be efficient from the learning perspective: any point which is far away from any of the classifiers will easily be classified as negative and will not contribute much to the loss function. In response, we propose multiple alternatives for sampling the negative points:

1. **Integral k-hop neighbours:** We sample the integral neighbours that are at an $L_1$ Distance of $k$ from $\mathbf{y}^*$. For small $k$, these form the hardest negatives as they are closest to the positive point. Note that it is possible for a few integral neighbours to be feasible *w.r.t.* the $m$ constraints, but they must have worse cost than the given ground truth $\mathbf{y}^*$. Such samples contribute towards learning the cost parameters $\theta_{\mathbf{c}}$.

2. **Project and sample:** We project the ground truth $\mathbf{y}^*$ on each of the $m$ hyperplanes and then randomly sample an integral neighbour of the projection, generating a total of $m$ negatives. Sampling probability in the $j^{th}$ dimension depends on the $j^{th}$ coordinate of the projection: if value of the $j^{th}$ coordinate is $r \notin \mathbb{Z}$, then we sample $floor(r)$ and $ceil(r)$ with probability $ceil(r) - r$ and $r - floor(r)$ respectively. If $r \in \mathbb{Z}$, then we sample $r$ with probability 1. Projection samples are

Table 1: Board accuracy and training time for different board sizes of symbolic and visual sudoku (for CombOptNet, "-" denotes time-out after 12 hours)

| | Symbolic Sudoku | | | | | | Visual Sudoku | | | | | |
|---|---|---|---|---|---|---|---|---|---|---|---|---|
| | Board Accuracy (%) | | | Training Time (min) | | | Board Accuracy (%) | | | Training Time (min) | | |
| | 4x4 | 6x6 | 9x9 | 4x4 | 6x6 | 9x9 | 4x4 | 6x6 | 9x9 | 4x4 | 6x6 | 9x9 |
| Neural (RRN) | **100.0** | 99.1 | 91.3 | 5 | 7 | 110 | **99.8** | 97.5 | 71.1 | 120 | 65 | 97 |
| CombOptNet | 0.0 | 0.0 | 0.0 | - | - | - | 0.0 | 0.0 | 0.0 | - | - | - |
| SATNet | 100.0 | 96.8 | 28.5 | 1 | 74 | 299 | 98.0 | 80.8 | 17.8 | 79 | 89 | 205 |
| ILP–Loss (Ours) | **100.0** | **100.0** | **100.0** | **1** | **2** | **52** | 99.7 | **98.8** | **98.3** | **3** | **11** | **92** |

close to the boundary of the currently learnt polyhedron, thus taking the training progress into account. Further, each hyperplane is likely to be assigned to a close-by negative due to projection sampling.
3. **Batch negatives:** We consider every other ground truth $\mathbf{y}^*_{s'}$, $s' \neq s$ in the minibatch as a potential negative sample for $\mathbf{y}^*_s$. This is particularly useful for learning the cost parameters $\theta_{\mathbf{c}}$ when the learnable constraints are the same for all the ground truth samples, such as in sudoku. In such cases, a batch–negative $\mathbf{y}^*_{s'}$ being a feasible point of the original ILP formulation, must always satisfy all of the $m$ learnable constraints of the original ILP in eq. (2). Hence, the only way for $\mathbf{y}^*_{s'}$ to be correctly classified as a negative for $\mathbf{y}^*_s$ is by violating the cost constraint in eq. (4), Learning cost parameters $\theta_{\mathbf{c}}$ that result in violation of the cost constraint for every batch–negative helps in ensuring that the ground truth $\mathbf{y}^*_s$ indeed has the minimum cost.
4. **Solver based:** Although our approach is motivated by the objective of avoiding solver calls, our framework is easily extensible to solver-based training by using the solution to the currently learnt ILP as a negative sample. This is useful when the underlying neural networks parameterizing $\mathbf{A}, \mathbf{b}$ or $\mathbf{c}$ are the bottleneck instead of the ILP solver. While we do not use solver negatives by default, we demonstrate their effectiveness in one of our experiments where the network parameterizing $\mathbf{c}$ indeed takes most of the computation time.

## 4 Experiments

The goal of our experiments is to evaluate the effectiveness and scalability of our proposed approach compared to the SOTA black-box solver based approach, i.e., CombOptNet [Paulus et al., 2021]. We experiment on 4 problems: symbolic and visual sudoku [Wang et al., 2019], and three problems from [Paulus et al., 2021]: random constraints, knapsack from sentence description and key-point matching. We also compare with an appropriately designed neural baseline for each of our datasets. To measure scalability, in each of the domains, we compare performance of each of the algorithms in terms of training time, as well as accuracy obtained, for varying problem sizes. For each of the algorithms, we report time till the epoch that achieves best val set accuracy and exclude time taken during validation. We kept a maximum time limit of 12 hours for training of each algorithm. We next describe details of each of these datasets, appropriate baselines, and our results. See the appendix for the details of the ILP solver used in our experiments, the hardware specifications, the hyper-parameters, and various other design choices.

### 4.1 Symbolic and Visual Sudoku

This task involves learning the rules of the sudoku puzzle. For symbolic sudoku, the input $x$ is a matrix of digits whereas for a visual sudoku, the $x$ is an image of a sudoku puzzle where a digit is replaced by a random MNIST image of the same class. A $k \times k$ sudoku puzzle can be viewed as an ILP with $4k^2$ equality constraints over $k^3$ binary variables [Bartlett et al., 2008]. For symbolic sudoku, each of the $k^2$ cells of the puzzle is represented by a $k$ dimensional binary vector which takes a value of 1 at $i^{th}$ dimension if and only if the cell is filled by digit $i$, resulting in a $k^3$ dimensional representation of the input $\mathbf{x} \in \mathbb{R}^n$ where $n = k^3$. On the other hand, for visual sudoku, each of the digit images in the $k^2$ cells are decoded (using a neural network) into a $k$ dimensional real vector, resulting in a $k^3$ dimensional **learnable** cost $\mathbf{c} \in \mathbb{R}^n$. The $k^3$ dimensional binary output vector (solution) $\mathbf{y}^*$ is created analogous to symbolic input. Our objective is to learn the constraint matrix $\mathbf{A} = f_{\mathbf{A}}(x; \theta_{\mathbf{A}}) = \theta_{\mathbf{A}}$ and vector $\mathbf{b} = f_{\mathbf{b}}(x; \theta_{\mathbf{b}}) = \theta_{\mathbf{b}}$, representing the linear constraints of sudoku. For symbolic sudoku, the cost vector $\mathbf{c} = -\mathbf{x}$ is known, where as for visual sudoku, the cost vector $\mathbf{c} = f_{\mathbf{c}}(x; \theta_{\mathbf{c}})$ is a function of the input image $x$, parameterized by $\theta_{\mathbf{c}}$ which also needs to be learned.

Table 2: Mean of the vector accuracy ($\mathbf{M}_\Theta(x) = \mathbf{y}^*$) and training time over the 10 random datasets in each setting of Random constraints. Number of learnable constraints is twice the number of ground truth constraints. See appendix for std. err. over the 10 runs.

| | | Vector Accuracy (%) | | | | Training Time (min) | | | |
|---|---|---|---|---|---|---|---|---|---|
| | | 1 | 2 | 4 | 8 | 1 | 2 | 4 | 8 |
| *Binary* | CombOptNet | 97.6 | 95.3 | 84.3 | 63.4 | 8.2 | 13.5 | 26.5 | 40.8 |
| | ILP–Loss (Ours) | **97.8** | **96.0** | **92.8** | **87.8** | **7.3** | **11.6** | **18.1** | **27.5** |
| *Dense* | CombOptNet | 89.3 | 74.8 | 34.3 | 2.0 | 9.9 | 16.8 | 24.7 | 48.2 |
| | ILP–Loss (Ours) | **96.6** | **86.3** | **74.0** | **41.5** | **7.3** | **15.6** | **17.6** | **20.6** |

**Dataset:** We experiment with 3 different datasets for $k = 4, 6$, and 9. We first build symbolic sudoku puzzles with $k$ digits, and use MNIST to convert a puzzle into an image for visual sudoku. For $9 \times 9$ sudokus, we use a standard dataset from Kaggle [Park, 2017], for $k = 4$ we use publically available data from Arcot and Kalluraya [2019], and for $k = 6$ we use the data generation process described in Nandwani et al. [2022]. We randomly select $10,000$ samples for training, and $1000$ samples for testing for each $k$. To generate the input images for visual-sudoku, we use the official train and test split of MNIST[Deng, 2012]. Digits in our train and test splits are randomly replaced by images in the MNIST train and test sets respectively, avoiding any leakage.

**Baselines:** Our neural baseline is Recurrent Relational Networks (RRN) [Palm et al., 2018]: a purely neural approach for reasoning based on Graph Neural Networks. We use the default set of parameters provided in their paper for training RRNs. We note that the RRN baseline uses additional information in the form of graph structure which is not available to our solver. We make this comparison to see how well we perform compared to one of the SOTA techniques for this problem. We also compare against another neuro-symbolic architecture using SATNet [Wang et al., 2019] as the reasoning layer. The neural component in all the neuro-symbolic methods is a CNN that decodes images into digits.

**Results:** Table 1 compares the performance and training time of our method against the different baselines described above. CompOptNet fails miserably on this problem, not being able to complete training for any of the sizes in the stipulated time. The purely neural model gives a decent performance and is competitive with ours for smaller board sizes. But for larger board size of $9 \times 9$, we beat RRN based model by a significant margin of about 25 points and 9 points in visual and symbolic sudoku, respectively. While SATNet performs comparable on smaller $4 \times 4$ sudoku puzzles, its performance degrades drastically to 17.8% for $9 \times 9$ board size on visual sudoku. See appendix for a comparison on the dataset used in Wang et al. [2019].

### 4.2 Random constraints

This is the synthetic dataset borrowed from Paulus et al.. The training data $\mathcal{D} = \{(\mathbf{c}_s, \mathbf{y}_s^*) | s \in \{1 \cdots S\}\}$ is created by first generating a random polyhedron $\mathcal{P} \subseteq \mathcal{Y}$ by sampling $m'$ hyperplanes $\{[\mathbf{a}_i' | \mathbf{b}_i'] \mid \mathbf{a}_i', \mathbf{b}_i' \in \mathbb{R}^n, i \in \{1 \cdots m'\}\}$ in an $n$ dimensional bounded continuous space $\mathcal{Y}$. A cost vector $\mathbf{c}_s \in \mathbb{R}^n$ is then randomly sampled and corresponding $\mathbf{y}_s^*$ is obtained as $\mathbf{y}_s^* = \arg\min_{\mathbf{z} \in \mathcal{P}} \mathbf{c}_s^T \mathbf{z}, \mathbf{z} \in \mathbb{Z}^n$. Objective is to learn a constraint matrix $\mathbf{A}$ and vector $\mathbf{b}$ such that for a ground truth $(\mathbf{c}, \mathbf{y}^*)$ pair, $\mathbf{y}^* = \arg\min_{\mathbf{z} \in \mathcal{Y}} \mathbf{c}^T \mathbf{z}, \mathbf{z} \in \mathbb{Z}^n$, and $\mathbf{A}\mathbf{z} + \mathbf{b} \geq \mathbf{0}$.

**Dataset:** Restricting $\mathcal{Y}$ to $[0, 1]^n$ and $[-5, 5]^n$ results in two variations, referred as 'binary' and 'dense' settings, respectively. For both of the output spaces, we experiment with four different settings in $n = 16$ dimensional space by varying the number of ground truth constraints as $m' = 1, 2, 4$, and 8. For each $m'$, we experiment with all the 10 datasets provided by Paulus et al. and report mean and standard error over the 10 models. The training data in each case consists of $1600$ $(\mathbf{c}, \mathbf{y})$ pairs and model performance is tested for 1000 cost vectors.

**Baseline:** Here we compare only against CombOptNet. A neural baseline (an MLP) is shown to perform badly for this symbolic problem in Paulus et al.. Hence we exclude this in our experiments.

**Results:** Table 2 presents the comparison of the two algorithms in terms of accuracy as well as time taken for training. While we perform marginally better than CombOptNet in terms of training time, our performance is significantly better, on almost all problems in the dense setting, and the larger problems in the binary setting. We are roughly 25 and 40 accuracy points better than CombOptNet

Table 3: Mean accuracy and train-time over 10 runs for various knapsack datasets with different number of items in each instance. For $N = 25, 30$, "-" represents timeout of 12 hours and we evaluate using the latest snapshot of the model obtained within 12 hours of total training. See appendix for std. err. over the 10 runs.

| | Vector Accuracy (mean in %) | | | | | Training Time (mean in min) | | | | |
|---|---|---|---|---|---|---|---|---|---|---|
| | 10 | 15 | 20 | 25 | 30 | 10 | 15 | 20 | 25 | 30 |
| CombOptNet | 63.2 | 48.2 | 30.1 | 2.6 | 0.0 | **41.0** | 61.4 | 153.0 | - | - |
| ILP–Loss (Ours) | **71.4** | **58.5** | **48.7** | **41.0** | **28.4** | 44.0 | **51.0** | **82.2** | **106.1** | **111.6** |

on the largest problem instance with 8 constraints in the binary and dense settings respectively. This result demonstrates that our approach is not only faster in terms of training time, but also results in better solutions compared to the baseline, validating the effectiveness of our approach.

### 4.3 Knapsack from Sentence Descriptions

This task, also borrowed from Paulus et al., is based on the classical NP-hard Knapsack problem. Each input consists of $N$ sentences describing the price and weight of $N$ items and the objective is to select a subset of items that maximizes their total price while keeping their total weight less than the knapsack's capacity $C$: $\arg\max \mathbf{p}^T \mathbf{z}$ s.t. $\mathbf{w}^T \mathbf{z} \leq C, \mathbf{z} \in \{0, 1\}^N$, where $\mathbf{p}, \mathbf{w} \in \mathbb{R}^N$ are the price and weight vectors, respectively. Each sentence has been converted into a $d = 4096$ dimensional dense embedding using Conneau et al. [2017], so that each input is $N \times d$ dimensional vector $\mathbf{x}$. The corresponding output is $\mathbf{y}^* \in \{0, 1\}^N$. The knapsack capacity $C$ is fixed for all instances in a dataset. The task is to learn the parameters $\theta_{\mathbf{c}}$, $\theta_{\mathbf{A}}$ and $\theta_{\mathbf{b}}$ of a neural network that extracts the cost and the constraints from $\mathbf{x}$. Note that here both the cost and the constraints need to be inferred from the input.

**Dataset:** The dataset in Paulus et al. has $4500$ train and $500$ test instances with fixed $N = 10$ items, extracted from a corpus containing $50,000$ sentences and their embeddings. In addition to experimenting with the original dataset, to demonstrate the scalability of our method, we also bootstrap new datasets with $N = 15, 20, 25$ and $30$, by randomly selecting sentences from the original corpus. Each bootstrapped dataset has $4500$ train and $500$ test instances.

**Results:** Table 3 presents a comparison between the training time and accuracy of our method against CombOptNet. We significantly outperform the baseline in terms of accuracy across all the datasets. For the smallest problem size with only 10 items, our training time is comparable to the baseline. This is expected as for small problems, ILP may not be the bottleneck, and the relative speedup obtained by solver free method gets offset by the increased number of iterations it may require to train. Our relative gain increases significantly with increasing problem size. CombOptNet fails to complete even a single epoch in the stipulated time and results in 0% accuracy on the largest problem.

### 4.4 Keypoint Matching

Here we experiment on a real world task of matching key points between two images of different objects of the same type Rolínek et al. [2020b]. The input $x$ consists of a pair of images $(I_1, I_2)$ of the same type along with a shuffled list of coordinates (pixels) of $k$ keypoints in both images. The task is to match the same keypoints in the two images with each other. The ground truth constraint enforces a bijection between the keypoints in the two images. The output $\mathbf{y}$ is represented as a $k \times k$ binary permutation matrix, *i.e.*, entries in a row or column should sum to 1. The cost $\mathbf{c}$ is a $k \times k$ sized vector parameterized by $\theta_{\mathbf{c}}$. The goal is to learn $\mathbf{A}$, $\mathbf{b}$ and $\theta_{\mathbf{c}}$.

**Dataset:** We experiment with image pairs in SPair-17k [Min et al., 2019] dataset used for the task of keypoint matching in [Rolínek et al., 2020b]. Since the neural ILP models can deal with a fixed size output space, we artificially create 4 datasets for 4, 5, 6 and 7 keypoints from the original dataset. While generating samples for $k$ keypoints dataset, we randomly sample any $k$ annotated pairs from input image pairs that have more than $k$ keypoints annotated. See appendix for details on dataset size.

**Baselines:** We compare against a strong neural baseline which is same as the backbone model parameterizing the $k^2$ dimensional cost vector in [Rolínek et al., 2020b]. It is trained by minimizing BCE loss between negative learnt cost ($-\mathbf{c}$) and target $\mathbf{y}^*$. We create an additional baseline by doing constrained inference with ground truth constraints and learnt cost ('Neural + CI') in table 4.

Table 4: Average point-wise accuracy and training times over 3 runs with different random seeds for varying # of keypoints. Neural + CI denotes ILP inference with known constraints over the cost learnt by neural model. See appendix for std. err. over the 3 runs.

| | Pointwise Accuracy (in %) | | | | Training Times (in min) | | | |
|---|---|---|---|---|---|---|---|---|
| | 4 | 5 | 6 | 7 | 4 | 5 | 6 | 7 |
| Neural | 80.88 | 78.04 | 75.39 | 73.49 | 148 | **37** | **3** | **40** |
| Neural + CI | 82.42 | 79.99 | 77.64 | 75.88 | 148 | **37** | 30 | **40** |
| CombOptNet | 83.86 | **81.43** | 78.88 | 76.85 | **41** | 67 | 144 | 279 |
| ILP–Loss (Ours) | 81.76 | 79.59 | 77.84 | 76.18 | 115 | 92 | 106 | 109 |
| ILP–Loss + Solver (Ours) | **84.64** | 81.27 | **79.51** | **78.59** | 43 | 73 | 99 | 174 |

**Results:** Table 4 presents the percentage of keypoints matched correctly by different models. In this experiment, the bottleneck *w.r.t.* time is the backbone neural model instead of the ILP solver. Therefore, we also experiment with solver based negatives in our method (ILP–Loss + Solver). While ILP–Loss using solver-free negatives performs somewhat worse than CombOptNet in terms of its accuracy, especially for smaller problem sizes, using solver based negatives helps ILP–Loss surpass CombOptNet in terms of both the accuracy and training efficiency for large problem sizes (and makes it comparable on others). This is because a solver based negative sample is guaranteed to be incorrectly classified (as positive) by the current hyperplanes, and hence provides a very strong training signal compared to solver-free negatives in ILP–Loss. We obtain a gain of up to 5 points over the Neural baseline, which is roughly double the gain obtained by Neural + CI.

## 5    Conclusion and Future Work

We have presented a solver–free framework for scalable training of a neural ILP architecture. Our method learns the linear constraints by viewing them as linear binary classifiers, separating the positive points (inside the polyhedron) from the negative points (outside the polyhedron). While given ground truth acts as positives, we propose multiple strategies for sampling negatives. A simple trick using the available ground truth outputs in the training data, converts the cost vector into a constraint, enabling us to learn the cost vector and constraints in a similar fashion.

Future work involves extending our method to *neural-ILP-neural* architectures, i.e., where the output of ILP is an input to a downstream neural model (see appendix for a detailed discussion). Second, a neural ILP model works with a fixed dimensional output space, even though the constraints for the same underlying problem are the same in first order logic, *e.g.*, constraints for $k \times k$ sudoku puzzles remain the same in first order irrespective of $k$. Creating neural ILP models that can parameterize the constraints on the basis of the size of the input (or learnt) cost vector is a potential direction for future work. Lastly, the inference time with the learnt constraints can be high, especially for large problems like $9 \times 9$ sudoku. In addition, the learnt constraints might not be interpretable even if the ground truth constraints are. In future, we would like to develop methods that distill the learnt constraints into a human-interpretable form, which may address both these limitations.

## Acknowledgements

We thank IIT Delhi HPC facility[4] for computational resources. We thank anonymous reviewers for their insightful comments that helped in further improving our paper. We also thank Ashish Chiplunkar, whose course on Mathematical Programming helped us gain insights into existing methods for ILP solving, and Daman Arora for highly stimulating discussions. Mausam is supported by grants from Google, Bloomberg, 1MG and Jai Gupta chair fellowship by IIT Delhi. Parag Singla was supported by the DARPA Explainable Artificial Intelligence (XAI) Program with number N66001-17-2-4032. Both Mausam and Parag are supported by IBM AI Horizon Networks (AIHN) grant and IBM SUR awards. Rishabh is supported by the CSE Research Acceleration fund of IIT Delhi. Any opinions, findings, conclusions or recommendations expressed in this paper are those of the authors and do not necessarily reflect the views or official policies, either expressed or implied, of the funding agencies.

---

[4]*http://supercomputing.iitd.ac.in*

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
