# Appendix

## 3 Differentiable ILP Loss

### 3.2 A Solver-free Framework

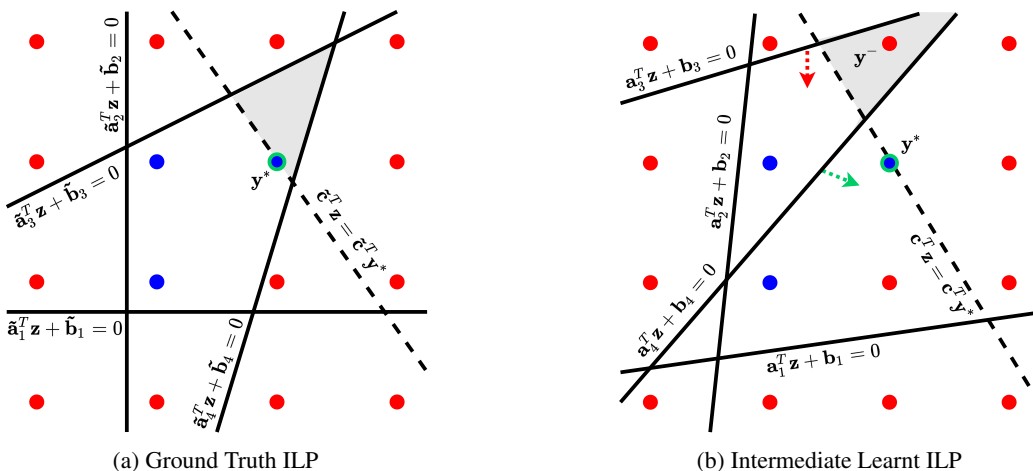

(a) Ground Truth ILP          (b) Intermediate Learnt ILP

Figure 1: An illustration of our framework. Figure on the left shows 4 ground truth constraints that need to be learnt. Blue dots are the only feasible integral points w.r.t. the 4 constraints. Shaded area containing only the dot with green border is the feasible region after we add the cost constraint (dashed line). Figure on the right shows an intermediate scenario while learning. The green-bordered dot (positive) is outside the intermediate $4^{th}$ constraint and the red dot (negative) is inside the intermediate $3^{rd}$ constraint. Positive and negative losses encourage the $4^{th}$ and the $3^{rd}$ hyperplanes to move in the direction shown by the green and red dotted arrows respectively.

In fig. 1, we consider a 2-dimensional ILP with 4 ground truth constraints $\tilde{\mathbf{a}_i}^T\mathbf{z}+\tilde{\mathbf{b}_i} \geq 0, i = 1, 2, 3, 4$, a cost vector $\tilde{\mathbf{c}}$, and target solution $\mathbf{y}^*$. The dots represent points in $\mathbb{Z}^2$. The solid lines represent hyperplanes (here lines) corresponding to the constraints, and the dashed line represents our cost-constraint. The shaded area in fig. 1a is the feasible region of the constraint satisfaction problem:

$$\arg\min_{\mathbf{z}} \mathbf{0}^T\mathbf{z} \text{ subject to } \tilde{\mathbf{A}}\mathbf{z} + \tilde{\mathbf{b}} \geq \mathbf{0} \; ; \; \tilde{\mathbf{c}}^T\mathbf{z} \leq \tilde{\mathbf{c}}^T\mathbf{y}^* \; ; \; \mathbf{z} \in \mathbb{Z}^2$$

Figure 1a shows the ground truth ILP. The signs of $\tilde{\mathbf{a}_i}$'s and $\tilde{\mathbf{b}}_i$'s are such that the closed polyhedron (here polygon) containing the blue points forms the feasible region $\tilde{\mathbf{A}}\mathbf{z} + \tilde{\mathbf{b}} \geq \mathbf{0}$. The green point is optimal w.r.t. the cost $\tilde{\mathbf{c}}$. The red points are infeasible, *i.e.*, $\tilde{\mathbf{A}}\mathbf{z} + \tilde{\mathbf{b}} < \mathbf{0}$ for red points. Note that the target solution $\mathbf{y}^*$ (point with green border) is the only integral point in the shaded region. The other blue points violate the cost constraint $\tilde{\mathbf{c}}^T\mathbf{z} \leq \tilde{\mathbf{c}}^T\mathbf{y}^*$, and the red points violate at least one of the four feasibility constraints $\tilde{\mathbf{a}_i}^T\mathbf{z} + \tilde{\mathbf{b}}_i \geq 0, i \in \{1 \cdots 4\}$.

Figure 1b shows a possible situation during learning. For simplicity, consider temperature close to zero ($\tau \approx 0$), so that only the closest hyperplane contributes to the negative loss for a negative sample. Also consider margins close to zero ($\mu^+ \approx 0, \mu^- \approx 0$). In fig. 1b, the point with the green border is outside the shaded region, whereas one red point is inside. The ground truth $\mathbf{y}^*$ is on the wrong side of only the fourth hyperplane, *i.e.*, $\mathbf{a}_4^T\mathbf{y}^* + b_4 < 0$, and hence it is the only contributor to the positive loss, *i.e.*, $L_+ = -\frac{1}{4}d(\mathbf{y}^*; [\mathbf{a}_4|\mathbf{b}_4])$, where $d(\mathbf{y}^*; [\mathbf{a}_4|\mathbf{b}_4]) = \frac{\mathbf{a}_4^T\mathbf{y}^*+\mathbf{b}_4}{|\mathbf{a}_4|} < 0$. The red negative point denoted as $\mathbf{y}^-$ (inside the shaded region) being closest to the third hyperplane contributes $d(\mathbf{y}^-; [\mathbf{a}_3|\mathbf{b}_3]) = \frac{\mathbf{a}_3^T\mathbf{y}^-+\mathbf{b}_3}{|\mathbf{a}_3|} > 0$ to the negative loss $L_-$. The green and red dotted arrows indicate the directions of movement of the constraint hyperplanes on weight update.

Table 5: Mean $\pm$ std err of the vector accuracy ($\mathbf{M}_\Theta(x) = \mathbf{y}^*$) and training time over the 10 random datasets in each setting of Random constraints. Number of learnable constraints is twice the number of ground truth constraints. CombOpt: CombOptNet

| | Vector Accuracy (%) | | | | Training Time (min) | | | |
|---|---|---|---|---|---|---|---|---|
| | 1 | 2 | 4 | 8 | 1 | 2 | 4 | 8 |
| Binary | | | | | | | | |
| CombOpt | $97.6 \pm 0.4$ | $95.3 \pm 0.5$ | $84.3 \pm 3.5$ | $63.4 \pm 4.0$ | $8.2 \pm 1.7$ | $13.5 \pm 1.6$ | $26.5 \pm 1.4$ | $40.8 \pm 4.0$ |
| ILP–Loss | $\mathbf{97.8 \pm 0.4}$ | $\mathbf{96.0 \pm 0.3}$ | $\mathbf{92.8 \pm 0.6}$ | $\mathbf{87.8 \pm 3.4}$ | $\mathbf{7.3 \pm 1.7}$ | $\mathbf{11.6 \pm 1.9}$ | $\mathbf{18.1 \pm 2.4}$ | $\mathbf{27.5 \pm 4.8}$ |
| Dense | | | | | | | | |
| CombOpt | $89.3 \pm 1.1$ | $74.8 \pm 1.9$ | $34.3 \pm 5.6$ | $2.0 \pm 0.6$ | $9.9 \pm 1.4$ | $16.8 \pm 1.3$ | $24.7 \pm 2.0$ | $48.2 \pm 2.3$ |
| ILP–Loss | $\mathbf{96.6 \pm 0.3}$ | $\mathbf{86.3 \pm 2.3}$ | $\mathbf{74.0 \pm 5.4}$ | $\mathbf{41.5 \pm 5.7}$ | $\mathbf{7.3 \pm 1.1}$ | $\mathbf{15.6 \pm 2.1}$ | $\mathbf{17.6 \pm 2.6}$ | $\mathbf{20.6 \pm 4.5}$ |

## 4 Experiments

**Details of the ILP Solver and the hardware used for experiments:** To solve the learnt ILPs, we use Gurobi ILP solver [Gurobi Optimization, LLC, 2022] available under 'Named-user academic license'. All our experiments were run on 11 GB 'GeForce GTX 1080 Ti' GPUs installed on a machine with 2.60GHz Intel(R) Xeon(R) Gold 6142 CPU. For each of the algorithms, we kept a maximum time limit of 12 hours for training.

### 4.1 Symbolic and Visual Sudoku

**Hyperparameters and other design choices:** (a) # of learnable constraints: We keep $m = (n+1)/2$ as the number of learnable equality constraints where $n = k^3$ is the number of binary variables. (b) Margin: We find that a margin of $0.01$ works well across domains and problem sizes. (c) Temperature: we start with a temperature $\tau = 1$ and anneal it by a factor of $0.1$ whenever the performance on a small held out set plateaus. (d) Early Stopping: we early stop the training based on validation set performance, with a timeout of 12 hours for each experiment. We bypass the validation bottleneck of solving the ILPs from scratch by providing the gold solutions as hints when invoking Gurobi. (e) Negative Sampling: For sampling neighbors, we select all the $n$ one hop neighbors, and an equal number of randomly selected 2,3 and 4 hop neighbors, resulting in a total of $4n$ neighbors. (f) Initialization: we initialize $\mathbf{a}_i$ from a standard Gaussian distribution for CombOptNet and our method.

**Comparison with SATNet on their dataset:** Wang et al. use a different set of $9 \times 9$ puzzles for training and testing sudoku and report 63.2% accuracy on visual sudoku, different from what it obtains on our dataset. Hence we trained both SATNet and our model on the dataset available on SATNet's Github repo. Interestingly, on their visual sudoku dataset which we believe to be easier (as shown by performance numbers), our run of SATNet achieves 71.0% board accuracy whereas our method achieves 98.3%.

### 4.2 Random Constraints

**Hyperparameters and other design choices:** We keep number of learnable constraints as twice the number of ground truth constraints, *i.e.*, $m = 2m'$, as Paulus et al. report best performance in most of the settings with it. Here we initialize $\mathbf{a}_i$ uniformly between $[-0.5, 0.5]$. Rest of the hyperparamters are set as in the case of sudoku.

**Results:** See table 5 for the standard error of the accuracy and training time over 10 random datasets for different number of ground truth constraints.

### 4.3 Knapsack from Sentence Descriptions

**Hyperparameters and other design choices:** Following Paulus et al., we use a two-layer MLP with hidden-dimension 512 to extract $\mathbf{c}$ and $\mathbf{A}$ from $N$ $d$-dimensional sentence embeddings, and keep $\mathbf{b} = 1$. Number of constraints $m$ is set to 4, a setting which achieves best performance in Paulus et al.. A notable difference is in the output layer of our MLP. Paulus et al. assume access to the ground truth price and weight range ($[10, 45]$ and $[15, 35]$ respectively), and use a sigmoid output

Table 6: Mean $\pm$ std. err. of accuracy and train-time over 10 runs for various knapsack datasets with different number of items in each instance. For $N = 25, 30$, *TO* represents timeout of 12 hours and we evaluate using the latest snapshot of the model obtained within 12 hours of total training.

CombOpt: CombOptNet

| | Vector Accuracy (mean $\pm$ std. err. in %) | | | | | Training Time (mean $\pm$ std. err. in min) | | | | |
|---|---|---|---|---|---|---|---|---|---|---|
| | 10 | 15 | 20 | 25 | 30 | 10 | 15 | 20 | 25 | 30 |
| CombOpt | 63.2$\pm$0.6 | 48.2$\pm$0.4 | 30.1$\pm$1.0 | 2.6$\pm$0.3 | 0.0$\pm$0.0 | **41.0$\pm$4.1** | 61.4$\pm$4.6 | 153.0$\pm$8.8 | *TO* | *TO* |
| ILP–Loss | **71.4$\pm$0.4** | **58.5$\pm$0.3** | **48.7$\pm$0.7** | **41.0$\pm$0.5** | **28.4$\pm$0.7** | 44.0$\pm$5.7 | **51.0$\pm$7.9** | **82.2$\pm$9.8** | **106.1$\pm$7.7** | **111.6$\pm$11.0** |

Table 7: Keypoint Matching: Number of train and test samples for datasets with different keypoints

| **Num Keypoints** | **4** | **5** | **6** | **7** |
|---|---|---|---|---|
| **#Test** | 10,474 | 9,308 | 7,910 | 6,580 |
| **#Train** | 43,916 | 37,790 | 31,782 | 26,312 |

non-linearity with suitable scale and shift to produce $\mathbf{A}$ and $\mathbf{c}$ in the correct range. We do the same for CombOptNet, but for ILP–Loss we simply use a linear activation at the output. We note that training CombOptNet with linear activation without access to the ground truth ranges gives poorer results.

**Results:** See table 6 for standard error over 10 runs with different random seeds for varying knapsack sizes.

### 4.4 Keypoint Matching

**Hyperparameters:** For each $k$, the number of learnable constraints is set to $2k$: same as the number of ground truth constraints. For keypoints 5,6 and 7, in addition to random initialization, we also experiment by initializing the backbone cost parameters $\theta_{\mathbf{c}}$ with the one obtained by training it on 4 keypoints and pick the one which obtains better accuracy on val set. This happens for all the methods for keypoints 6 and 7.

For ILP–Loss with only solver and batch negatives (ILP–Loss + Sol.) , we start with a temperature $\tau = 0.1$ and anneal it by a factor of $0.5$ whenever performance on a small validation set plateaus. For ILP–Loss with only solver–free negatives, we start with a temperature $\tau = 0.5$ and anneal it by a factor of $0.2$ at $10^{th}$ and $30^{th}$ epoch. As done in Paulus et al. [2021], we initialize $\mathbf{a}_i$ uniformly between $[-0.5, 0.5]$. Rest of the hyperparmeters are same as those used for sudoku.

**Dataset details:** See table 7 for the number of train and test samples in the four datasets created for $4, 5, 6,$ and 7 keypoints.

**Results:** See table 8 for the standard error of point-wise accuracy and training times over 3 runs with different random seeds for varying number of keypoints.

## 5 Future Work

**Discussion on training Neural-ILP-Neural architectures:** In the current formulation, availability of the solution to the ground truth ILP is important for our solver-free approach to work. Specifically, it is required to: 1.) convert the constrained optimization problem to a constraint satisfaction problem by including the cost-constraint eq. (4), and 2.) to calculate the positive loss eq. (6). However, in a Neural-ILP-Neural architecture, the intermediate supervision for only the Neural-ILP part (*i.e.*, solution of the ground truth ILP) is not available.

On the other hand, solver based methods such as CombOptNet, do not require access to the solution of the ground-truth ILP. Instead, they rely on the solution of the current intermediate ILP (during learning) to compute the gradients of the loss and thus their approach is not solver free. We note that even though in princple they can train Neural-ILP-Neural architectures, their experiments are only in the Neural-ILP settings.

Extending our current work for Neural-ILP-Neural architectures is an important direction of future work. One plausible approach could be to train an auxiliary inverse network that converts a given

Table 8: Mean ± std error of point-wise accuracy and training times over 3 runs with random seeds for varying number of keypoints. Neural+CI denotes ILP inference with known constraints over the cost learnt by neural model.

| | Pointwise Accuracy (mean ± std err in %) | | | | Training Times (mean ± std. err. in min) | | | |
|---|---|---|---|---|---|---|---|---|
| | 4 | 5 | 6 | 7 | 4 | 5 | 6 | 7 |
| Neural | 80.88±0.87 | 78.04±0.40 | 75.39±0.50 | 73.49±0.55 | 148±26 | **37±12** | **30±9** | **40±13** |
| Neural + CI | 82.42±0.55 | 79.99±0.16 | 77.64±0.25 | 75.88±0.43 | 148±26 | **37±12** | **30±9** | **40±13** |
| CombOptNet | 83.86±0.62 | **81.43±0.49** | 78.88±0.65 | 76.85±0.54 | **41±15** | 67±8 | 144±31 | 279±39 |
| ILP–Loss | 81.76±1.71 | 79.59±0.18 | 77.84±0.36 | 76.18±0.06 | 115±13 | 92±3 | 106±1 | 109±5 |
| ILP–Loss + Sol. | **84.64±0.62** | 81.27±1.12 | **79.51±0.53** | **78.59±0.55** | 43±12 | 73±10 | 99±9 | 174±25 |

output of Neural-ILP-Neural architecture to a predicted symbolic target of Neural-ILP component. This predicted target can be used as a proxy to the ground truth solution of the Neural-ILP part. Similar ideas of using an inverse network have been explored in [Agarwal et al., 2021], albeit under the setting where ILP is known and only the neural encoder and decoder needs to be learnt.