# OpenReview forum: "A Solver-free Framework for Scalable Learning in Neural ILP Architectures"
_NeurIPS.cc/2022/Conference — NeurIPS 2022 Accept_

### Official Review · Reviewer_ZncX · 2022-07-11

**Rating:** 7
**Confidence:** 5
**Soundness:** 3 good
**Presentation:** 3 good
**Contribution:** 3 good

**Summary:**

This paper introduces a solver-free approach for learning the constraint and cost parameters of an integer linear program from data. This means that the method does not rely on an expensive invocation of the ILP solver on the forward pass.
Instead, it relies on the available ground-truth solutions to train a set of soft max-margin classifiers corresponding to the ILP constraints and one additional constraint representing the cost.
The classifiers are trained such that a ground truth solution is classified positively by all classifiers, and any other integer point (sampled to be difficult to classify) is classified negatively by at least one of the classifiers.

The presented method (ILP-Loss) is evaluated on three experiments, including visual sudoku, random synthetic constraints, and a real-world keypoint matching benchmark. The main competing method is CombOptNet [1], which relies on invoking the ILP solver on every forward pass.

In the visual Sudoku experiment, the authors also include an additional SOTA baseline (RRN [2]) which is based on a graph neural network. ILP-Loss outperforms the RRN baseline on the largest board instances and performs comparably on the smaller ones while training significantly faster. CombOptNet times out after a very long training time, presumably because the ILP solver invocations dramatically slow down the training.
In the random constraints experiment, ILP-Loss again outperforms CombOptNet in terms of accuracy and runtime.
Finally, in the keypoint matching experiment, ILP-Loss is shown to achieve comparable performance to CombOptNet at a similar runtime, as the computational bottleneck of this experiment is in the neural network backbone.

[1]: Anselm Paulus, Michal Rolínek, Vít Musil, Brandon Amos, and Georg Martius. Comboptnet: Fit the right np-hard problem by learning integer programming constraints. ICML 2021.

[2]: Rasmus Berg Palm, Ulrich Paquet, and Ole Winther. Recurrent relational networks. NeurIPS 2018.

**Questions:**

See the suggestions in the quality section.

**Limitations:**

The authors discussed the limitation that currently a neural-ILP-neural architecture is not possible with the presented method. It would be great to include a short discussion of why this is not possible, and contrast it with some of the existing work which is applicable to this setting.

**Strengths And Weaknesses:**

### Originality
The authors present a method for learning the constraint and cost coefficients of an ILP as multiple soft max-margin classifiers. While this approach has already been used to learn a classifying convex polytope from data (as also mentioned by the authors in the abstract), this is to the best of my knowledge the first time that the cost terms have been learned as an additional classifier, thereby extending the setting to cover integer linear programs.

### Quality
Overall the quality of this paper is good.
However, there is one major weakness in my opinion, which is a missing evaluation of the setting in which there is no single set of constraints describing the whole dataset, but in which instead the constraints depend on the input $x$. The presentation of the method suggests that this is doable (line 157), however, none of the experiments validates this. I am particularly surprised by the missing comparison as such an experiment has already been included in [1], and while all the other experiments from [1] were repeated this is the only one missing. I would ask the authors to include a comparison on this experiment, as I believe in any case (the presented method performing good or bad), the community would gain important insights. I will raise my rating of this paper if the authors include the missing comparison.

In addition, I would like to also point out that the papers [1,3,4], which are cited as neural-ILP architectures (line 44), are in fact capable of (and designed as) neural-ILP-neural architectures. This provides important context, especially for the experimental comparison to [1], and therefore should be corrected and discussed.

### Clarity
The clarity of the paper is good, it is easy to read and understand the introduced method.

### Significance
I think this paper is significant, as it appears to be the currently best method for learning the constraint parameters of a neural-ILP architecture. The performance on Visual Sudoku outperforming the current state-of-the-art method is impressive and demonstrates the capabilities of the presented method.



[3]: Marin Vlastelica Pogancic, Anselm Paulus, Vít Musil, Georg Martius, and Michal Rolínek. Differentiation of blackbox combinatorial solvers. ICLR 2020.

[4]: Quentin Berthet, Mathieu Blondel, Olivier Teboul, Marco Cuturi, Jean-Philippe Vert, and Francis Bach. Learning with differentiable pertubed optimizers. NeurIPS 2020.

---

> ### Author Response · Authors · 2022-08-02
> **Official Response to Reviewer ZncX**
>
> Dear reviewer ZncX,
> We thank you for a very careful read of our paper and suggestions to make it better. Below we address your concerns:
>
>
> ### 1. Experiment on Knapsack dataset (cost dependent on input)
> As correctly pointed by you, our formulation is generic enough to incorporate the case when both the cost and the constraints are a function of the input. There is no specific reason for not including an experiment on the knapsack dataset in the original draft.
>
> As suggested by you, we trained the Neural-ILP  architecture using both ILPLoss and CombOptNet on the knapsack dataset as provided in the CombOptNet paper.
>
> Since the original dataset has only 10 items in a Knapsack, it leads to an ILP in only 10 dimensional binary space. To further test the scalability of our solver-free approach, we bootstrapped new knapsack problems with 15, 20, 25, and 30 items using the 50,000 sentences and their embeddings available in the original dataset, resulting in 4 additional datasets with 15, 20, 25, and 30 dimensional ILPs respectively. We report the performance of both the training methods on the original as well as these four new datasets in the table below. For both the methods, we keep a total of 4 learnable constraints. This corresponds to the setting which achieved the best performance reported in CombOptNet paper.
>
> **Knapsack Test Accuracy (In pct)**: For size 25 and 30, CombOptNet could not finish even a single epoch in 12 hours. We report the numbers computed using the latest checkpoint of the model within 12 hours.
> |     Accuracy(In %)  |  **10**  |  **15**  | **20** |  **25**  | **30** |
> |----------------|:--------:|:--------:|:------:|:--------:|:------:|
> | **CombOptNet** |     64.8 |       46 |   27.8 |     4.2     |    0.0 |
> |    **ILPLoss** | **72.8** | **56.6** | **47.0** | **37.2** | **27.0** |
>
> **Train-times (In mins)**: For size 25 and 30, TO denotes that the training timeout of 12 hours was reached for CombOptNet.
>
> | Train-times(min) | **10** | **15** | **20** | **25** | **30** |
> |--------------------------|:------:|:------:|:------:|:------:|:------:|
> |           **CombOptNet** |   46.3 |   91.4 |  156.2 |    TO |    TO |
> |              **ILPLoss** |   **20.1** |   **43.8** |  **112.1** |  **161.6** |  **102.2** |
>
> We would also like to point out an important implementation difference. CombOptNet assumes access to the ground truth range of the prices and weights of the items in the knapsack and accordingly maps the output of the neural encoder to a pre-specified range. On the other hand, we do not make any such assumption and instead rely on our method to learn these constraints on its own.
>
>
>
> ### 2. [1,3,4] capable of training Neural-ILP-Neural architectures [1,3,4], but incorrectly cited as Neural-ILP (line 44)
>
> We agree that the training techniques proposed in [1,3,4] are generic and can indeed be used to train Neural-ILP-Neural.  Since our technique focuses on, (or is rather limited to) training Neural-ILP architectures, we didn't want to confuse the reader in the Introduction section itself. Nevertheless, we will clarify it in the updated draft. But interestingly, none of the experiments in CombOptNet paper demonstrate its capability to train a Neural-ILP-Neural architecture and after including experiments on Knapsack, we are now comparing against the CombOptNet approach in all the settings reported in the CombOptNet paper.
>
>
> ### 3. Discussion on why is ILPLoss not applicable for Neural-ILP-Neural architectures?
>
> Our formulation of ILPLoss requires the solution of the ground truth ILP for two things:
> 1. To convert the constrained optimization problem to a constraint satisfaction problem by including the cost-constraint (line#193 and eqn 4).
> 2. To calculate the positive loss (eqn 6).
>
> Availability of the solution to the ground truth ILP is important for our solver-free apporach to work, at least in the current formulation. However, in a Neural-ILP-Neural architecture, the intermediate output of only the Neural-ILP part (i.e. solution of the ground truth ILP) is not available. Extending our current approach for Neural-ILP-Neural architectures is an important direction of future work. One plausible approach could be to  train an auxiliary inverse network that converts a given output of Neural-ILP-Neural architecture to a predicted symbolic target of Neural-ILP. This predicted target can be used as a proxy of ground truth solution to Neural-ILP.
>
> On the other hand, solver based methods such as CombOptNet, do not require access to the solution of the ground-truth ILP. Instead, they rely on the solution of the current intermediate ILP (during learning) to compute the gradients of the loss and this solution is obtained by solving the current ILP. We note that even though in principle they can train Neural-ILP-Neural architectures, the experiments in [1,3,4] are only on Neural-ILP settings.
>
> Please let us know if any of your concerns are still unaddressed, and we will be happy to respond. Thanks!

---

> > ### Comment · Reviewer_ZncX · 2022-08-06
> > **Answer to Rebuttal**
> >
> > I thank the authors for including the suggested additional experiment and addressing my remaining concerns regarding the paper. I will update my rating of the paper once the revised version with the discussed changes is submitted.

---

> > > ### Author Response · Authors · 2022-08-06
> > > **Reply to Reviewer ZncX**
> > >
> > > Dear reviewer ZncX,
> > >
> > > Thank you for your response. We are actively working on a revised manuscript incorporating the suggested changes and additional experiments. We will be uploading it soon.
> > >
> > > Best,
> > > Authors

---

> > > > ### Author Response · Authors · 2022-08-08
> > > > **Reply to Reviewer ZncX**
> > > >
> > > > Dear reviewer ZncX,
> > > >
> > > > We have now uploaded a rebuttal revision of our paper and have also posted a common comment describing all the changes in the revised manuscript.  We hope that all of your concerns have been appropriately addressed in the revision. If not, please let us know, and we will address them.
> > > >
> > > > Thanks!

---

### Official Review · Reviewer_HE5s · 2022-07-11

**Rating:** 7
**Confidence:** 3
**Soundness:** 3 good
**Presentation:** 3 good
**Contribution:** 3 good

**Summary:**

This work proposes a novel method for end-to-end training integer linear programming. The proposed method converts the ILP problem into a satisfiability problem, in which all constraints are transformed into inequalities. During the learning stage, the parameters of the inequality constraints can be learned via gradient descent given positive examples and randomly sampled negative examples. Experiments show that the proposed approach is effective and scalable. However, I am not an expert in integer programming, so I cannot judge the novelty of this work.

**Questions:**

1. Is it possible to recover the inequalities from the learned parameters?
2. How does your method perform when comparing to SATNet?


**Limitations:**

Yes

**Strengths And Weaknesses:**

Pros:
1. The proposed approach is more scalable than previous methods, and the performance is good.
2. This paper has covered extensive related works and provides a detailed discussion about them.
3. The authors also propose several effective methods for sampling negative examples.

Cons:
1. Compared to the solver-based method, the proposed approach abandons the explicit symbolic representation, so it will be difficult to generalize outside of the training examples, e.g., training on 4x4 sudoku and test on 9x9 sudoku.
2. Converting the problem to a classification problem increases the sample complexity.

---

> ### Author Response · Authors · 2022-08-02
> **Official response to Reviewer HE5s**
>
> Dear reviewer HE5s,
>
> We thank you for your review and a pointer to a future work. Below we address your concerns and answer your questions.
>
> ### 1. Generalize outside of the training examples, e.g., training on 4x4 sudoku and test on 9x9 sudoku.
>
> This is a very interesting direction for future work where instead of learning a fixed number of constraints over a fixed dimensional space, we instead generate variable number of constraints in a variable number of dimensions based on the input size. This is akin to representing and learning constraints in first order logic. To the best of our knowledge, there are only a couple of works addressing this, albeit in a purely neural setting [Dong et al., 2019, Nandwani et al., 2022], and  no existing work addresses it in a neuro-symbolic architecture.
>
>
> ### 2. Converting the problem to a classification problem increases the sample complexity.
>
> Please note that in our experiments we trained our models using the same number of training samples as used in CombOptNet and achieved better performance, especially on difficult synthetic datasets with 8 ground truth constraints. Hence, we observe empirically that accuracy of our method is better than the baseline CombOptNet for the same number of training samples. To further evaluate the sample complexity of ILPLoss as compared to CombOptNet, we re-trained on the synthetic datasets with 50% of the training data and observed similar gains in our method over the baseline.
>
> We note that ILPLoss (our method) may need additional negative samples, but they are not required as part of the training data, and are instead generated by our algorithm. Hence, they should not count towards increasing the sampling complexity.
>
> ### 3. Is it possible to recover the inequalities from the learned parameters?
>
> Indeed! Once the parameters $\theta_A, \theta_b$ are learnt, we know the exact inequalities for an input $x$ as $f_A(\theta_A,x)z + f_b(\theta_b,x) \ge 0$. Please see line#156-161.
>
>
> ### 4. How does your method perform when comparing to SATNet?
>
> Thank you for suggesting this comparison.
>
> We did not report it earlier because SATNet uses a very important additional information in the form of input mask (indicating which cells are empty in the visual sudoku board). Topan et al. 2021 and Chang et al. 2020 report that SATNet achieves 0% test accuracy if the input mask is not provided. On the other hand, the Neural-ILP formulation does not assume access to input mask.
>
> Nevertheless, as suggested by you, we additionally ran experiments using SATNet. The table below compares our performance with SATNet on our training and test datasets for all the three sizes: 4x4, 6x6 and 9x9. While SATNet performs comparable on smaller 4x4 sudoku puzzles, its performance degrades drastically to just 17.8% on our dataset for 9x9 size.
>
> |              | Board Accuracy (In pct) |       |       | Train-Time <br>(in min.) |       |       |
> |--------------|:--------------------------:|:-----:|:-----:|:---------------------------:|:-----:|:-----:|
> | **Board Size**   |            **4 x 4**           | **6 x 6**  | **9 x 9** |            **4 x 4**            | **6 x 6**   | **9 x 9** |
> | Neural (RRN) |                       99.4 |  **97.6** |  74.4 |                         120 |    65 |    97 |
> | CombOptNet   |                        0.0 |   0.0 |   0.0 |                         720 |   720 |   720 |
> | SATNET       |                       98.0 |  80.8 |  17.8 |                          79 |    89 |   205 |
> | Ours         |                       **99.8** |  94.7 |  **99.0** |                           **2** |     **5** |    **45** |
>
>
> We would like to note that the SATNet paper uses a different set of puzzles for training and testing sudoku. Therefore we went a step ahead and trained both SATNET and our model on the dataset in official SATNet's repo. Even on their dataset which we believe to be easy (as shown by performance numbers), our run of SATNet achieved only 71% board accuracy (SATNet paper reported their performance to be 63.2%), whereas our method achieved 98.3%. This clearly demonstrates the benefits of a Neural-ILP architecture over Neural-SATNet that uses a MAXSAT formulation (with a low rank SDP relaxation) for the symbolic reasoning.
>
> Please let us know if we any of your concerns are still unaddressed and we will be happy to respond.
>
> Thanks!
>
> [Chang et al. 2020]: Assessing SATNet’s Ability to Solve the Symbol Grounding Problem, NeurIPS 2020
>
> [Dong et al. 2019]: Neural logic machines, ICLR 2019
>
> [Nandwani et al. 2022]:Neural models for output-space invariance in combinatorial problems, ICLR 2022
>
> [Topan et al. 2021]: Techniques for Symbol Grounding with SATNet, NeurIPS 2021

---

> > ### Comment · Reviewer_HE5s · 2022-08-08
> > **Comment to the response**
> >
> > Thank you for your feedback, it has addressed my questions.

---

### Official Review · Reviewer_ZzdC · 2022-07-11

**Rating:** 7
**Confidence:** 2
**Soundness:** 3 good
**Presentation:** 3 good
**Contribution:** 3 good

**Summary:**

This paper presents a method  to reduce the cost of adding a final ILP stage to a NN..

**Questions:**

Tab 3 should have running times?
RRNs: the extra info  is? Is it used by the RRN resukts in the paper?
Diid you consider the other benchmarks in the RRN paper?
3.2 How does your system handle misclassified data? Can you introduce slack?
3.3 and 4,1 Why 3.3, given 4.1?
3.4 what is the difference?

**Limitations:**

The main limitations addressed in this paper is scalability; it would be nice to have more application ex.

**Strengths And Weaknesses:**

The paper explains its ideas clearly, It essentially transforms a linear ILP classifier into a continuous derivable problem my moving using distances to the hyperplanes. One may argue that this is not an ILP problem any longer, but it still stems from the original formulation, The results on sudoku loook good, supporting the idea,

Weaknesses:
- How easy it is to encode problems in LP?The examples are rater limited,
- The other  results are not as supportive, namely Table 3 (what happened to the run-times?)

---

> ### Author Response · Authors · 2022-08-02
> **Official Response to Reviewer ZzDc (Part 1)**
>
> Dear reviewer ZzdC,
>
> Thank you for your review and some interesting questions. Below we make an attempt to answer them:
>
> ### 1. How easy it is to encode problems in LP?
> Our formulation of ILPLoss is generic and can in principle encode an LP problem. But we believe making it perform well on LP problems may require looking into some additional challenges which we highlight below and addressing them is a direction for future work.
>
> In the case of an ILP, while generating the negative samples for the classifiers, we make use of the integerality of solutions, (e.g., we sample k hop neighbors). In the case of an LP, the given targets (optimal $y$'s) of a parameterized LP can still be used as positive data samples for training the unknown linear cost and constraint classifiers,  but it is unclear how to sample negatives without integerality assumption: even $y + \epsilon$, for any arbitrarily small $\epsilon$, is also a valid negative sample.
>
> In addition, in the case of ILPs, because of discrete output space, the same underlying problem can be represented using multiple ILPs. In other words, many ILPs can have the same feasible set, and learning any one of them would suffice. For example,
> $$0 \le x_1 \le 5 ; \  0 \le x_2 \le 5 ; \ x_1,x_2 \in \mathbb{Z}$$ and
> $$0 \le x_1 \le 5+\delta_1 ;\  0 \le x_2 \le 5 + \delta_2; \ x_1,x_2 \in \mathbb{Z}$$ represent the same integeral space for any $0 \le \delta_1, \delta_2 < 1$. We believe this lack of flexibility in terms of which constraints need to be learned for representing an LP, may pose additional challenges that may affect learnability.
>
>
> ### 2. Run-times in Keypoint Matching (Table 3)
>
> We do not observe any train-time speedup in the Keypoint Matching experiment. This is because the bottleneck in the corresponding Neural-ILP architecture is not the ILP solver but the backbone perception model comprising of a VGGNet and a Graph Neural Network (90-95% of the time in our method is spent in the neural backbone). As discussed in the paper (see line#409-413), due to the higher number of iterations required by our method, we at times observe worse train times than CombOptNet.
>
> Nevertheless, as suggested by the reviewer, we report the exact train times in the table below.
>
>
> Train-time (in minutes) for different datasets with different number of keypoints:
> |            |   4 |  5 |  6 |   7 |
> |------------|----:|---:|---:|----:|
> | CombOptNet |  89 | 98 | 50 |  57 |
> | ILPLoss (ours)    | 155 | 77 | 86 | 100 |
> | Neural     | 108 | 12 | 17 |  41 |
>
>
> ### 3. Extra info in RRN and other benchmarks
>
> **Extra information:** RRN is a recurrent variant of a message passing Graph Neural Network that takes a graph as input. For sudoku, the graph nodes are sudoku cells (total 81 nodes for 9x9 sudoku). When adding edges, the RRN paper [Palm et al., 2018] assume that in sudoku value of a node is constrained by values of other nodes in the same row, column, and block, and hence they connect each node with all other nodes in its row, column, and block. Hence, the edges capture the structure of the underlying problem. On the other hand, no such additional information is provided when we formulate sudoku as an Integer Linear Program in our method.
>
>
> **Other benchmarks:** The other benchmarks in the RRN paper can not be formulated as an ILP and hence are not relevant to our paper. They are:  bAbI [Weston et al., 2015] and Pretty-CLEVR. bAbI is a text-based QA task in which a question along with some facts in natural language is provided to a model, and the objective is to select one of the 177 words from full bAbI vocabulary as an answer. Pretty-CLEVR involves natural language reasoning questions over a scene consisting of 8 objects of different colors and shapes.
>
>
> ### 4. How does your system handle misclassified data? Can you introduce slack variables?
>
> There are two possible scenarios:
> 1. The underlying data generation process solves an exact ILP, but there may be noise in the available training data.
> 2. The underlying data generation process itself solves an ILP with soft constraints (modeled using slack variables) so that the given solutions in the clean training data can violate the ground truth hard constraints.
>
> In the first scenario, the objective is to learn the ground truth cost and constraints, and in the second scenario, one also needs to learn the penalty for the continuous slack variables, in addition to the cost for the integer variables.
>
> In both the above scenarios, the given targets in the training data can be misclassified by the ground truth constraints. Our ILPLoss formulation can handle the 1st scenario as it involves *soft-margin* loss (eqn. 5 to 9), similar to soft margin SVM formulation which is known to learn robust classifiers. The introduction of continuous slack variables in the 2nd scenario will convert it into a Mixed Integer Linear Program (MILP), handling which is an interesting direction for future work.
>
> *Continued in the next comment...*

---

> > ### Author Response · Authors · 2022-08-02
> > **Official Response to Reviewer ZzDc (Part 2)**
> >
> > ### 5. Why 3.3, given 4.1? 3.4 what is the difference?
> >
> > Section 3.3 (Negative Sampling) is part of our formulation that needs negative samples to compute the loss. Section 3.4 (Implementation Details) enumerates the details required for computing the loss and are common to all the experiments, while section 4.1 contains the details relevant to only the sudoku experiment.  Please let us know if we have not understood your question and if it is still unanswered.
> >
> > ### 6. More Experiments
> >
> > We would like to note that in addition to visual sudoku, we have now experimented with all the tasks considered in the CombOptNet paper. See the results on the Knapsack dataset in our response to reviewer ZncX. In addition, we also included neural baselines wherever relevant, i.e. in sudoku (RRN) and in Keypoint Matching (neural VggNet+GNN without constraints and VggNet + GNN with constrained inference). As requested by reviewer HE5s, for sudoku, we have now experimented with SATNet baseline as well and report much better performance on standard 9 x 9 sudokus. See response to HE5s for a comparison.
> >
> >
> > Please let us know if any of your concerns are still not addressed, and we will be happy to respond.
> >
> > Thanks!

---

### Author Response · Authors · 2022-08-08
**Summary of changes in the Rebuttal Revision**

We again thank all the reviewers for their insightful comments on our manuscript.  The list below enumerates all the changes that we have made in the rebuttal revision:

1. Added a comparison against SATNet in visual sudoku.
2. Added a new experiment on Knapsack from Sentence Description.
3. Added train-times for different methods in the keypoint matching experiment.
4. Added a discussion on training of Neural-ILP-Neural architectures.
5. Added a footnote clarifying that some of the existing cited works can, in principle, train Neural-ILP-Neural architectures.

To incorporate additional experiments within the page limit of 9 pages, we decided to move implementation details and hyper-parameter choices to the appendix. If any reviewer has any other suggestions about it, please let us know, and we will be happy to incorporate them. For better readability, the newly added text is in blue, and the text we moved (mainly to the appendix at the end) is in red.

Please let us know if any of your comments are still not addressed in the updated manuscript, and we will be happy to address them.


Thanks!

---

### Meta-Review · Area_Chair_ztz2 · 2022-08-26

**Recommendation:** Accept
**Confidence:** Certain

**Metareview:**


The paper presents the first solver-free training approach for learning neural integer linear programs. The idea is to encode within the loss that the final polyhedron separates the positives (all constraints satisfied) from the negatives (negatives (at least one violated constraint or a suboptimal cost value), via a soft-margin formulation. Compared to the Neural ILP baselines, this turns out to be faster without sacrificing accuracy. All reviewers agree that this is solid work. I fully agree.

**Award:**

No

---

### Decision · Program_Chairs · 2022-09-14

Accept